# Ataxia Rating Scales: Content Analysis by Linking to the International Classification of Functioning, Disability and Health

**DOI:** 10.3390/healthcare10122459

**Published:** 2022-12-05

**Authors:** Mohammad Etoom, Alhadi M. Jahan, Alia Alghwiri, Francesco Lena, Nicola Modugno

**Affiliations:** 1Physical Therapy Department, Aqaba University of Technology, Aqaba 77110, Jordan; 2School of Rehabilitation Sciences, University of Ottawa, Ottawa, ON K1Y 4W7, Canada; 3Department of Physiotherapy, College of Medical Technology, Misrata 51, Libya; 4Department of Physiotherapy, School of Rehabilitation Sciences, University of Jordan, Amman 11942, Jordan; 5Department of Medicine and Health, University of Molise, 86100 Campobasso, Italy; 6IRCCS INM Neuromed, 86077 Pozzilli, Italy

**Keywords:** patient outcome assessment, ICF linking, content analysis, outcomes measurement, treatment outcome, rehabilitation

## Abstract

Ataxia management is mainly based on rehabilitation, symptomatic management, and functional improvement. Therefore, it is important to comprehensively assess ataxic symptoms and their impact on function. Recently, the movement disorders society recommended four generic ataxia rating scales: scale for assessment and rating of ataxia (SARA), international cooperative ataxia rating scales, Friedreich’s ataxia rating scale (FARS), and unified multiple system atrophy rating scale (UMSARS). The aim of the study was to analyze and compare the content of the recommended ataxia rating scales by linking them to the international classification of functioning, disability and health (ICF). A total of 125 meaningful concepts from 93 items of the four included scales were linked to 57 different ICF categories. The ICF categories were distributed in body structure (*n* = 8), body function (*n* = 26), activity and participation (*n* = 20), and environmental factors (*n* = 3) components. UMSARS and FARS were the only ones that have addressed the body structure or environmental factors component. The content analysis of ataxia rating scales would help clinicians and researchers select the most appropriate scale and understand ataxic symptoms and their impact on function. It seems that SARA is the optimal scale for rapid assessment of ataxia or in busy clinical settings. UMSARS or FARS are more appropriate for the investigating the impact of ataxia on overall health, and monitoring ataxia progression and disability.

## 1. Introduction

Ataxia is a group of impairments of the ability to perform coordinated movements due to neurological disorders that can be sporadic or inherited [1]. It has broad and heterogeneous effects on mobility, speech, psychological status, and quality of life [2]. Management of ataxia is mainly based on rehabilitation, symptomatic management, and functional improvement [3]. Therefore, it is important to comprehensively assess ataxic symptoms and their impact on function.

The clinical assessment of ataxia can be performed by a set of ataxia rating scales [4]. Recently, the movement disorders society recommended four generic ataxia rating scales based on using the scales in different ataxic populations and psychometric proprieties [4]. The scales are: scale for assessment and rating of ataxia (SARA) [5], international cooperative ataxia rating scales (ICARS) [6], Friedreich’s ataxia rating scale (FARS) [7], and unified multiple system atrophy rating scale (UMSARS) [8]. These ataxia rating scales showed adequate psychometric proprieties represented with feasibility, acceptability, consistency, and reproducibility. Moreover, the scales can be used in different ataxic populations [4]. 

The recommended ataxia rating scales show similarities and differences in the covered ataxia aspects. For example, the SARA mainly assesses ataxia through motor performance, while ICARS is concerned with assessing of oculomotor disorders. The autonomic functions and bulbar dysfunctions were assessed by UMSARS and FARS, respectively. Accordingly, it is essential to analyze and compare the content of ataxia rating scales. One way to assess and compare the content of rating scales is by linking them to the International Classification of Functioning, disability and health (ICF). The Linking to ICF would provide a structured description and comparison of the content of each ataxia rating scale. Therefore, the aim of this study was to analyze and compare the content of the recommended ataxia rating scales by linking them to the ICF. 

## 2. Materials and Methods

### 2.1. Selection of Ataxia Rating Scales

We have selected four generic ataxia clinical rating scales that were recommended by movement disorders society: SARA, ICARS, FARS, and UMSARS [4]. 

### 2.2. Description of Ataxia Rating Scales

(1)SARA has eight items to assess motor performance, speech disturbance, coordination, and limb kinetic functions. The application time is 14.2 ± 7.5 min [5].(2)ICARS consists of 19 items, divided into posture and gait disturbance, limb kinetic function, speech disorders, and oculomotor disorders. The application time is 21.3 ± 7 min [9].(3)FARS has 36 items distributed in 4 domains: (I) functional staging of ataxia; (II) activity of daily life; (III) neurological assessment of bulbar, upper and lower limbs, peripheral nerve, and upright stability/gait functions; and (IV) quantitative timed activities—PATA rate, nine-hole pegboard, and timed 25-foot walk test. It needs more than 30 min to be administrated [7].(4)UMSARS has 30 items comprising four parts, including a historical review of disease-related impairments, motor examination, autonomic examination, and the global disability scale. The application time is 30–45 min [10].

### 2.3. ICF

The ICF was developed by the world health organization in 2001 to define different aspects of functioning, disability, and health. It aims to provide a common language for disability and health that enables a better understanding of health and health-related states [11].

In the ICF codes, the letters b, s, d, and e, refer to the components of the classification: body structure (s), body function (b), activity and participation (d), or environmental factors (e). The component letter is followed by a numeric code starting with the first level [chapter], second level, third level, and rarely fourth level. The following example illustrated an ICF category in the activity and participation component:

d Activity and participation [component level]

d4 Mobility [first level/ chapter] 

d415 Maintaining a body position [second level]

d4154 Maintaining a standing position [third level]

### 2.4. Procedure of ICF Linking

The linking processes were conducted using updated linking guidelines and refinements [12,13] in two phases. All items and their responses in the included scales were surveyed to identify the meaningful concepts for each item independently by the two raters (M.E and A.J) in the first phase. More than one meaningful concept may be acknowledged for an item. A list of meaningful concepts was discussed to be used in the second phases.

In the second phase, each meaningful concept was linked to one or more ICF categories by detecting the most suitable component, first level (chapter), second level, and third level. The ICF categories were independently selected by the two raters. In case of disagreement between raters, a third rater who has an extensive experience in ICF (A.A) [14,15,16] was referred to resolve the disagreement and provided the rationale for the most appropriate code.

### 2.5. Interrater Rating Agreement

Cohen’s kappa statistic was used to evaluate the interrater agreement between the two raters on identification of meaningful concepts, and ICF category linkage process. Cohen’s kappa value ranges from 0 to 1, in which 0 indicates no agreement and 1 indicates perfect agreement [17]. Interrater agreement was estimated for meaningful concepts and component level, then first/chapter level, second, and third level category. The analyses were done at a 95% confidence interval.

### 2.6. ICF Linking Indicators

The following ICF linking indicators were calculated to compare scales and their relationship to ICF [18]:

#### 2.6.1. Measure to ICF Linkage

Is the Percentage of Items in a Scale that Can Be Linked to the ICF Category = The Number of Items Linked to At Least 1 ICF Code/Total Number of Items on the Measure × 100%.

#### 2.6.2. Measure of Linking to Unique ICF Codes

Percentage of Items in a Scale that Could Be Linked to Unique and Unrepeated ICF Code = Number of Items that Are Linked to Unique ICF Code/Total Number of Items on the Scale × 100% 

## 3. Results

### 3.1. Meaningful Concepts

The most identified meaningful concept was “movement coordination” (20 concepts, 16%), followed by “tremor” and “walking” (8 concepts, 6.4% each).

### 3.2. ICF Linking Results

The meaningful concepts were linked to 57 different ICF categories. Two meaningful concepts were not linked to ICF. They were “stage of disability” from the global disability scale item in UMSARS, and from functional staging of ataxia item in FARS, and they were assigned to “nd-dis”.

The 57 different ICF categories were distributed in body structure component (*n* = 8), body function component (*n* = 26), activity and participation component (*n* = 20), and environmental factors component (*n* = 3). Table 1 summarizes the total number of items, meaningful concepts, ICF components and categories, and ICF indicator results. Table 2 shows the linked ICF categories among the four ataxia rating scales.

#### 3.2.1. Representation of Body Structure

The 8 ICF categories in the body structures component are included in two chapters: 

Chapter 3: Structures involved in voice and speech,

Chapter 7: Structures related to movement.

The body structure component was covered only by UMSARS and FARS (Table 2, Figure 1).

#### 3.2.2. Representation of Body Function

The 26 ICF categories in the body function component are included in five chapters: 

Chapter 2: Sensory functions and pain,

Chapter 4: Functions of the cardiovascular, hematological, immunological, and respiratory systems,

Chapter 5: Functions of the digestive, metabolic and endocrine systems,

Chapter 6: Genitourinary and reproductive functions,

Chapter 7: Neuromusculoskeletal and movement-related functions.

All the included scales included concepts that are referred to chapter 7. The ICF category “b 7602 coordination of voluntary movements” was linked with all scales. “b 7651 tremor”, and “b 770 gait pattern functions” were linked with three out of the four included scales. Chapter 2 was covered by ICARS and UMSARS, while chapters 4, 5, and 6 were covered by UMSARS and FARS (Table 2, Figure 1).

#### 3.2.3. Representation of Activity and Participation

The 20 ICF categories activities and participation are included in five chapters:

Chapter 1: Learning and applying knowledge, 

Chapter 3: Communication,

Chapter 4: Mobility,

Chapter 5: Self-care,

Chapter 6: Domestic life.

All the included scales have concepts that referred to chapters 3 and 4. “d 4500 walking short distances” was linked with all scales. “d 4509 walking, unspecified”, and “d 465 moving around using equipment” were linked to three out of the four included scales. Chapters 1, 5, and 6 were covered by UMSARS and FARS only (Table 2, Figure 1). 

#### 3.2.4. Representation of Environmental Factors

The 3 ICF environmental factors are included in two chapters:

Chapter 1 Products and technology,

Chapter 3 Support and relationships.

The environmental factors component was covered by only UMSARS and FARS (Table 2, Figure 1). 

### 3.3. Content Comparison

UMSARS and FARS were the only scales that have addressed the body structure or environmental factors component. In the body function component, chapters 4, 5, and 6 were covered by UMSARS and FARS only, as well as chapters 1, 5, and 6 in the activity and participation component. Furthermore, UMSARS and FARS have the highest representation in chapter 7 in the body function component (Figure 1). 

The measure of linking to unique ICF codes percentage ranged from 36.8% to 76.6% (Table 1). The ICARS has the lowest measure of linking to unique ICF codes percentage.

### 3.4. Unspecified-ICF Categories

There were three unspecified-ICF categories as follows:(1)“b 7159 Stability of joint functions, unspecified” that was linked to “Falling” concept in UMSARS and FARS.(2)“d 4509 Walking, unspecified” that was linked for “changing walking direction” or “tandem walking” concepts in all included scales.(3)“d 599 Self-care, unspecified” that was linked to “hygiene care” concept in FARS.

### 3.5. Agreement between Authors

Table 3 shows the Kappa agreement at a 95% confidence interval between the two raters. The estimated kappa values for the scales ranged from 0.67 to 0.84 for meaningful concepts identification, and from 0.67 to 1 for ICF categories. The overall Kappa reflects substantial to perfect agreement between raters in the linkage process.

## 4. Discussion

The current study provides a content analysis and comparison of ataxia rating scales. The selection of recommended ataxia rating scales with sound psychometric proprieties allows for comparison of the content. There is a set of ICF linking studies for vestibular symptoms [16], fatigue [15], fear of falling [19], pain [20], and quality of life outcomes [21]. The majority of studies have linked self-reported questionnaires and included all potential outcomes despite the psychometric evaluations that limit the comparability and choosing of appropriate scales [19]. The current study included performance-based ataxia measures. The included ataxia rating measures show similarities and differences in the covered ataxia aspects as well as the variances in the meaning of the terms and response scale. Therefore, we think that it is important to analyze the content of the included scales. To the best of our knowledge, this is the first study that analyze the content of generic ataxia rating scales. Content analysis by linking to ICF is an important step for content validity. The ICF provides a valuable framework and classification method for coding components of scales by linking them to ICF components and categories [13]. 

The included ataxia rating scales focus mainly on body function and activity and participation components. Chapter 7: Neuromusculoskeletal and movement-related functions in body function component with “b 755 Involuntary movement reaction functions”, “b 7600 Control of simple voluntary movements”, “b 7602 Coordination of voluntary movements”, “b 7650 Involuntary movement functions”, “b 7651 Tremor”, and “b 770 Gait pattern functions” categories were the more frequent. In the activity and participation component, Chapter 4 Mobility; “d 4153 Maintaining a sitting position”, “d 4154 Maintaining a standing position”, “d 4500 Walking short distances”, “d 4509 Walking, unspecified”, and “d 465 Moving around using equipment” categories were the more frequent. Ataxia is characterized by an inability to perform coordinated movements, and wide range of ataxic involuntary movement, and abnormal gait ataxic patterns [22]. Gait deficits are typically the presenting sign of ataxia [1]. The included scales have a low representation of the body structure and environmental factors component. It is crucial to assess body structure impairments in ataxia such as postural deformities. The involuntary movements and ataxic pattern lead to muscle imbalance and postural deformities and, therefore, falls [23]. The fall rating was assessed only in UMSARS and FARS. 

The comparison between outcomes was based on the ICF indicators, and the covered ICF components and categories especially at the first/chapter level. There is a clear superiority of UMSARS and FARS in terms of the covered ICF categories. Moreover, these scales have higher unique meaningful concepts and ICF codes. These scales assess non-ataxia symptoms such as peripheral nervous system, autonomic, and bulbar functions. SARA and ICARS have fewer items and need less time to perform [24]. The redundant items of ICARS did not represent more content diversity, ICARS has repeated concepts and ICF categories more than other scales that reflect in low measure of linking to unique ICF codes, Table 1. Development of a short-form of ICARS based on psychometric evaluations and content would be beneficial. We think that SARA is an optimal scale for rapid assessment of ataxia, or in busy clinical settings. UMSARS, or FARS are more appropriate for assess impact of ataxia on health, functioning, and disability. They are more beneficial for monitoring ataxia progression, the activity of daily life, and independency.

A few limitations of this study should be noted. First, the inclusion of generic ataxia rating scales and exclusion of functional scales, quality of life, and other-related ataxic outcomes was a limitation. The ICF does not fully cover all meaningful concepts. For example, the meaningful concepts “Changing walking direction”, “tandem gait”, and “falling” were linked to multiple ICF codes to cover the meaningful concept. Furthermore, we noted some complexity in items as the majority of items had more than one meaningful concept. This is because of the wide responses for items. For example, item 1 with its responses in the ICARS have four meaningful concepts. Also, there were undefined ICF categories. The current study linked four ataxia rating scales to ICF. Future studies developing an ICF core set for ataxia are required. An ICF ataxia core set should consider the patient’s perspective and different healthcare professionals including neurologists and rehabilitation professionals, such as physiatrists, and physical, occupational, and speech therapists. 

## 5. Conclusions

The current study analyzed and compared the content of recommended ataxia rating scales. This would help clinicians and researchers in selecting most appropriate scale and understanding the ataxic symptoms and their impact on function. It seems that SARA is an optimal scale for rapid assessment of ataxia, or in busy clinical settings. UMSARS or FARS are more appropriate for assessing the impact of ataxia on overall health, monitoring ataxia progression, the activity of daily life, and disability.

## Figures and Tables

**Figure 1 healthcare-10-02459-f001:**
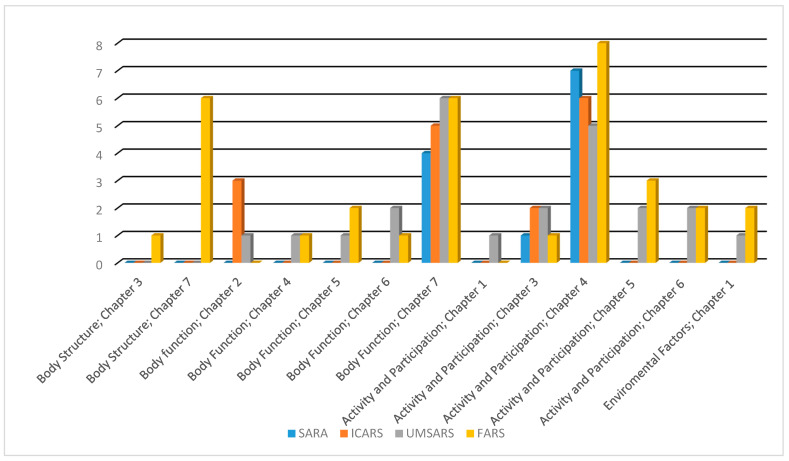
Comparison between the linked ICF categories of ataxia rating scales in terms of the covered chapters among ICF components. FARS, Friedreich’s ataxia rating scale; SARA scale for assessment and rating ataxia; UMSARS, unified multiple system atrophy rating scale.

**Table 1 healthcare-10-02459-t001:** Summary of the frequencies of the meaningful concepts, ICF components and categories, and ICF indicators for the included ataxia rating scale. ICARS, international cooperative ataxia rating scales; FARS, Friedreich’s ataxia rating scale; SARA, scale for assessment and rating ataxia; UMSARS, unified multiple system atrophy rating scale.

Scale	SARA	ICARS	UMSARS	FARS
Numbers of items	8	19	30	36
Number of concepts (Different)	14 (10)	26 (15)	39 (28)	45 (34)
Concept not linked to ICF	0	0	1	1
Total ICF categories (Unique)	15 (13)	33 (16)	40 (30)	56 (39)
Body structure	0	0	1	7
Body function	8	21	22	23
Activity and participation	7	12	16	22
Environmental factors	0	0	2	4
ICF indicators:				
Measure to ICF linkage	100%	100%	96.7%	97.2%
Measure of linking to unique ICF codes	6/8 (75%)	9/19 (47.4%)	23/30 (76.7%)	24/36 (66.7%)

**Table 2 healthcare-10-02459-t002:** Frequencies of ICF categories among the four ataxia rating scales. ICARS, international cooperative ataxia rating scales; FARS, Friedreich’s ataxia rating scale; SARA scale for assessment and rating ataxia; UMSARS, unified multiple system atrophy rating scale.

ICF Component	ICF Category	SARA	ICARS	UMSARS	FARS
Body structure	Chapter 3: Structures involved in voice and speech				
	s 3203 Tongue	0	0	0	1
	Chapter 7: Structures related to movement				
	s 7104 Muscles of head and neck region	0	0	0	1
	s 73002 Muscles of upper arm	0	0	0	1
	s 73012 Muscles of forearm	0	0	0	1
	s 73022 Muscles of hand	0	0	0	1
	s 75002 Muscles of thigh	0	0	0	1
	s 75012 Muscles of lower leg	0	0	0	1
	s 760 Structure of trunk	0	0	1	0
Body Function	Chapter 2: Sensory functions and pain				
	b 2152 Functions of external muscles of the eye	0	3	1	0
	Chapter 4: Functions of the cardiovascular, hematological, immunological and respiratory systems				
	b 4201 Decreased blood pressure	0	0	2	0
	b 450 Additional respiratory functions	0	0	0	1
	Chapter 5: Functions of the digestive, metabolic and endocrine systems				
	b 51050 Oral swallowing	0	0	1	1
	b 5253 Faecal continence	0	0	0	1
	Chapter 6: Genitourinary and reproductive functions				
	b 6202 Urinary continence	0	0	1	1
	b 640 Sexual functions	0	0	1	0
	Chapter 7: Neuromusculoskeletal and movement-related functions				
	b 7152 Stability of joints generalized	0	0	1	0
	b 7159 Stability of joint functions, unspecified	0	0	1	1
	b 7300 Power of isolated muscles and muscle groups	0	0	0	3
	b 7350 Tone of isolated muscles and muscle groups	0	0	1	0
	b 7500 Stretch motor reflex	0	0	0	1
	b 755 Involuntary movement reaction functions	2	0	0	0
	b 7600 Control of simple voluntary movements	1	0	4	2
	b 7602 Coordination of voluntary movements	3	5	0	5
	b 7650 Involuntary movement functions	0	1	0	0
	b 7651 Tremor	1	5	2	0
	b 770 Gait pattern functions	0	1	1	1
Activity and participation	Chapter 1: Learning and applying knowledge				
	d 170 Writing	0	0	1	0
	Chapter 3: Communication				
	d 330 Speaking	1	1	2	3
	d 3350 Producing body language	0	0	1	0
	d 3352 Producing drawings and photographs	0	1	0	0
	Chapter 4: Mobility				
	d 4103 Sitting	1	0	1	0
	d 4104 Standing	1	1	1	0
	d 4153 Maintaining a sitting position	1	2	0	2
	d 4154 Maintaining a standing position	1	1	1	4
	d 4301 Carrying in the hands	0	0	0	1
	d 4400 Picking up	0	0	0	1
	d 4403 Releasing	0	0	0	1
	d 4500 Walking short distances	1	2	2	2
	d 4509 Walking, unspecified	1	2	0	2
	d 465 Moving around using equipment	1	1	2	1
	Chapter 5: Self-care				
	d 510 Washing oneself	0	0	1	0
	d 5400 Putting on clothes	0	0	1	1
	d 5402 Putting on footwear	0	0	0	1
	d 599 Self-care, unspecified	0	0	0	1
	Chapter 6: Domestic life				
	d 6300 Preparing simple meals	0	0	1	1
	d 6600 Assisting others with self-care	0	0	1	1
Environmental Factors	Chapter 1 Products and technology				
	e 1151 Assistive products and technology for personal use in daily living	0	0	1	1
	e 1201 Assistive products and technology for personal indoor and outdoor mobility and transportation	0	0	0	2
	Chapter 3 Support and relationships				
	e 340 Personal care providers and personal assistants	0	0	1	1

**Table 3 healthcare-10-02459-t003:** Kappa agreement (95% confidence interval) at meaningful concepts, and ICF categories for the ataxia rating scales. ICARS, international cooperative ataxia rating scales; FARS, Friedreich’s ataxia rating scale; SARA, scale for assessment and rating ataxia; UMSARS, unified multiple system atrophy rating scale.

Scale	SARA	ICARS	UMSARS	FARS	Overall
Meaningful concepts	0.75 (0.486 to 1.000)	0.668 (0.372 to 0.965)	0.732 (0.511 to 0.952)	0.835 (0.655 to 1.000)	0.762 (0.622 to 0.903)
ICF Component	1	0.878 (0.713 to 1)	0.781 (0.578 to 0.985)	0.842 (0.669 to 1)	0.843 (0.749 to 0.937)
First/chapter level	0.732(0.387 to 1)	0.824 (0.632 to 1)	0.733 (0.514 to 0.953)	0.796 (0.605 to 0.988)	0.747 (0.625 to 0.868)
Second Level	0.714 (0.348 to 1)	0.721 (0.493 to 0.949)	0.689 (0.459 to 0.918)	0.714 (0.499 to 0.929)	0.705 (0.577 to 0.832)
Third Level	0.717 (0.462 to 0.972)	0.666 (0.421 to 0.911)	0.645 (0.405 to 0.885)	0.675 (0.453 to 0.898)	0.644 (0.504 to 0.783)

## Data Availability

The data presented in this study are available on request from the corresponding author.

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
