# Peer review of "Ataxia Rating Scales: Content Analysis by Linking to the International Classification of Functioning, Disability and Health"

_healthcare, 2022, doi:10.3390/healthcare10122459_

Round 1
Reviewer 1 Report
The authors have compared 4 commonly used ataxia rating scales and analysed them by comparing with the ICF. THey conclude that each scale has advantages and disadvantages - SARA being quick and clinically useful, whereas UMSARS and FARS are more useful in the research context.
This paper provides a useful addition to the literature. It is generally well presented, concise and clear. It makes no unreasonable assumptions or conclusions.
Reviewer 2 Report
Dear Authors, dear Editors,
Thank you for the opportunity to review your manuscript.
The authors performed the content analysis of ataxia rating scales by linking them to the ICF with the goal of helping researchers and clinicians with the appropriate selection of ataxia rating scales.
In the past, a set of ICF linking studies for vestibular symptoms or pain was published, but a majority were linked to the self-reported questionnaires and it makes me sense.
In this study, authors chose ataxia scales SARA and ICARS which are clinical scales based on semi-quantitative assessments of cerebellar ataxia and more complex FARS and UMSARS.
SARA is used for patients with SCA or others in which cerebellar symptomatology dominates the clinical picture. FARS is used for FRDA patients because this exam includes neurological signs that specifically reflect neural substrates in FRDA – except the cerebellum - pyramidal tracts and peripheral neurons. UMSARS was designed for patients with multiple system atrophy who not only have cerebellar but also extrapyramidal, pyramidal symptomatology and autonomic failure.
There is a discussion about SARA (quicker) and FARS (more complex and more accurate but time-consuming) applications in FRDA patients but this study won't settle that.
In conclusion, it is unclear how linking to the ICF could help in choosing the appropriate scale and for which kind of patients it is suitable.
Therefore I can´t recommend this paper for publication.
Reviewer 3 Report
This is an interesting paper examining the content of 4 ataxia scales based on ICF categories. It importantly shows the areas covered and not covered across these scales, as well as possible redundancy within and across scales. There are a few small edits that are needed.
1) Lines 84 and 88: The authors describe the ICF coding as having 1 digit per level, but in the example, it looks like the 2nd level has 2 digits (15). This inconsistency should be corrected.
2) line 121 - this is not a complete sentence
3) line 247 - "there" should be "they"
4) line 256 - there should be a space between 1 and with
5) line 257 and 263 - "scale" should be "scales"
6) line 263 - the "and" at the end of this line should be deleted
